# Telling the boiling frog what he needs to know: why climate change risks should be plotted as probability over time

Simon Sharpe[1]

[1]Institute for Innovation and Public Purpose, University College London, Gower Street, London, WC1E 6BT, United Kingdom

*Correspondence to*: Simon Sharpe (simonsharpe79@hotmail.com)

**Abstract.**  Humanity's situation with respect to climate change is sometimes compared to that of a frog in a slowly-boiling pot of water, meaning that change will happen too gradually for us to appreciate the likelihood of catastrophe and act before it is too late.  I argue that the scientific community is not yet telling the boiling frog what he needs to know.  I use a review of the figures included in two reports of the Inter-governmental Panel on Climate Change to show that much of the climate science

communicated to policymakers is presented in the form of projections of what is most likely to occur, as a function of time (equivalent to: 'in 5 minutes' time, the water you are sitting in will be two degrees warmer').  I argue from first principles that a more appropriate means of assessing and communicating the risks of climate change would be to produce assessments of the likelihood of crossing non-arbitrary thresholds of impact, as a function of time (equivalent to: 'the probability of you being boiled to death will be 1% in five minutes' time, rising to 100% in twenty minutes' time if you don't jump out of the pot').  This

would be consistent with approaches to risk assessment in fields such as insurance, engineering, and health and safety. Importantly, it would ensure decision-makers were informed of the biggest risks, and hence of the strongest reasons to act.  I suggest ways in which the science community could contribute to promoting this approach, taking into account its inherent need for cross-disciplinary research and for engagement with decision-makers before the research is conducted, instead of afterwards.

## 1 Introduction and argument from first principles

As the conceptual framework of 'risk assessment' is increasingly applied to climate change, the need to consider low probability, high impact risks ('tail risks') is often pointed out (Weitzman, 2011; IPCC, 2014a).  What is not so often mentioned is that this principle is a subsidiary of a more general principle, which is perhaps taken to be self-evident: that a risk assessment should consider the biggest risks.  In the case of a climate change risk assessment, how should we ensure that it does so?


If the magnitude of a risk is a function of probability and impact, then a risk assessment must consider three fundamental variables: probability, impact and time.  To be sure of identifying the biggest risks, all three variables must be explored fully. But to fully explore any two of them, the third must be held constant.  So the question is which choice of constant will lead to the fullest assessment of the risks.

If a risk is unchanging over time (at least to a rough approximation), then the answer is simple: hold time as constant by fixing a duration of interest, and then plot impact against probability. An earthquake risk graph as shown in Fig. 1 is such an example. It shows the full range of probabilities and impacts, from which the biggest risks can be understood. The time period is arbitrary, but changing it would not provide any significant further information.

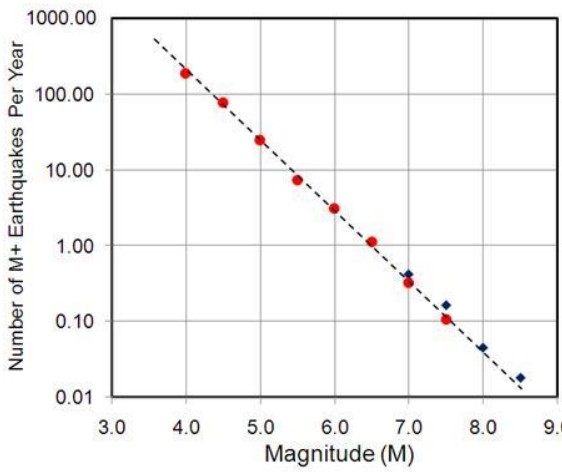

**Figure 1: Example of probability / impact graph for a risk unchanging over time: plot of frequency / magnitude of earthquakes in the Chile area** (Braile, 2010)

For risks that change over time, the choice is not so obvious. If time is held constant at a fixed point, then the full range of probabilities and impacts can be explored at that point, but bigger risks that may occur at different times will not be visible. If probability is held constant and impact plotted against time, then bigger risks may be omitted either because they correspond to a probability other than that which has been chosen, or because they would occur at a later time than is shown on the x axis. (Shaded bands illustrating uncertainty in impact can bring a broader range of risks into view, but still provide no guarantee that the biggest risks will be visible.) Similarly, if impact is held constant and probability plotted against time, then bigger risks may be omitted if they have larger impacts or occur at later times. These differences are illustrated in the three different plots of global temperature increase, probability and time shown in Fig. 2.

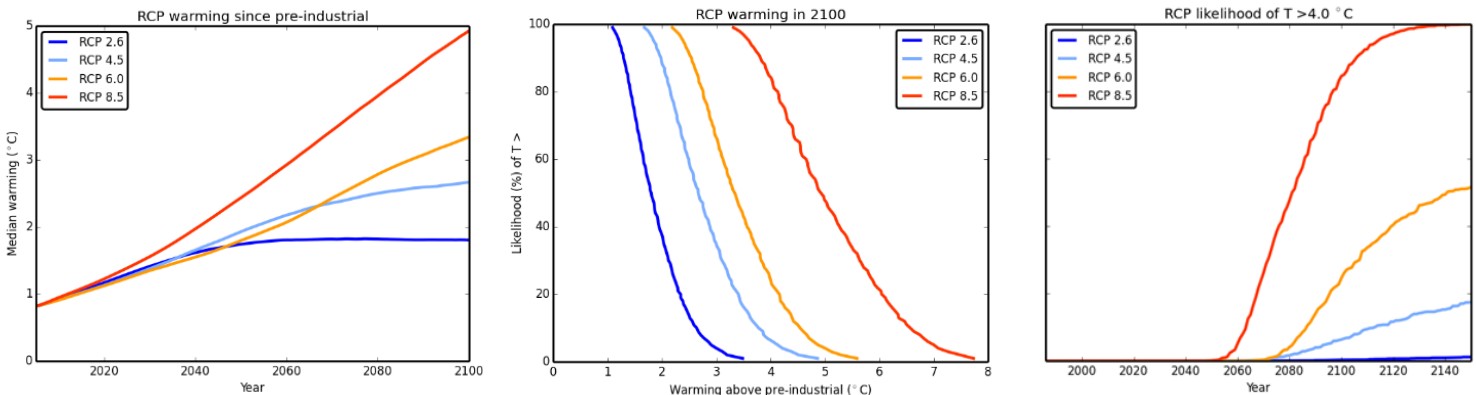

**Figure 2: Three configurations of probability, impact and time, where temperature increase = 'impact': a) impact / time, (fixed probability); b) probability / impact, (fixed time); c) probability / time, (fixed impact);** (Lowe and Bernie, 2015)

The difference between these three approaches lies in the relative arbitrariness of their fixed points. Fixing time makes little sense because while there is an obvious starting point (the present), there is no obvious end-point or discontinuity. Any fixed future point in time (e.g. the year 2100) is arbitrary. Probability has some interesting values, such as 0.5: the point at which something becomes more likely than not. But it is not clear that any particular value has special relevance for a risk assessment:
the biggest risks could occur at 1%, 5%, or 63%. Furthermore, the fact that the range of probability is bounded at both ends – by 0 and 1 – makes it particularly well-suited to being one of our axes.

Impact, by contrast, may well have some fixed points that are not arbitrary, but highly meaningful. This can be seen in examples of regulations for the structural integrity of buildings in earthquakes, the capital reserve requirements for insurance
firms, and the health and safety standards for people at work, which set maximum tolerable probabilities for building collapse, insurance firm insolvency, and worker death respectively. In each of these cases, the chosen probability is arbitrary, but the chosen fixed point of impact is not. For the building, insurance firm, or worker, the impacts chosen represent 'worst case' outcomes, beyond which no greater impact would be possible. On the range of possible severities of impact, these points represent discontinuities. Where such discontinuities can be identified, it may be most useful for a risk assessment to plot the
probability of encountering them as a function of time.

To illustrate the relevance of this for risk assessment, consider the proverbial frog in a slow-boiling pot of water. If the frog asks his science adviser for advice, and is told that in five minutes, the water will be warmer by 2°C plus or minus a degree or two (illustrated with an impact over time graph), he may decide there is no compelling reason for him to get out. If instead he
asks first what is the worst that could happen, and then how likely this is, his adviser will tell him that he could be boiled to death, and that while the probability of this is low within the next five minutes, it is rising over time, and at some point it will become more likely than not. Presented with the graph of probability of boiling as a function of time, the policy conclusion for the frog will be relatively clear.

Climate change has no single, obvious, 'boiling frog' scenario. There is no temperature threshold within which we are safe, and beyond which we are all cooked. Still, there is no reason why a similar approach could not be taken to assess a range of climate change risks. For example, Sherwood and Huber (2010) estimated that climatic conditions exceeding human physiological tolerance for heat stress would occur in parts of the world when temperatures rose 7°C above the late-20th century average.[1] This may be compared, albeit not exactly, with the probability of exceeding 7°C above pre-industrial, estimated as
a function of time for RCP8.5 by Jason Lowe and Dan Bernie, using a simple climate model to represent the climate sensitivity probability distribution function of the CMIP5 model ensemble (Fig. 3).

---

[1] Analysis by Dr. Tord Kjellstrom, Professor Alistair Woodward, Dr. Laila Gohar, Professor Jason Lowe, Dr. Bruno Lemke, Lauren Lines, Dr. David Briggs, Dr. Chris Freyberg, Dr. Matthias Otto, and Dr. Olivia Hyatt in King et al., 2015 (pp57-63) showed that in some regions this threshold may begin to be crossed at significantly lower values of global average temperature increase.

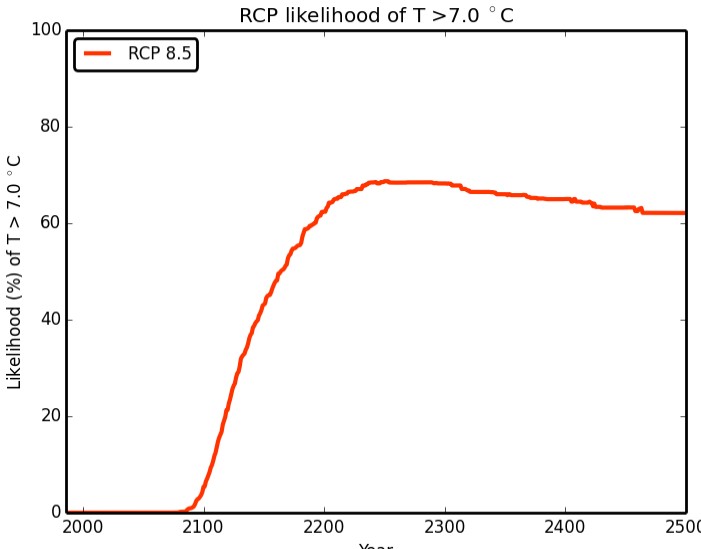

**Figure 3: A 'boiling frog' example: the probability of global mean temperature exceeding 7°C above pre-industrial, as a function of time, for RCP8.5** (Lowe and Bernie, 2015)

While the probability of exceeding this threshold of temperature rise is low within the arbitrarily-defined time period of this century, it appears to rise rapidly thereafter, until it becomes more likely than not. Shown this way, a risk that might be assumed in the short-term to be negligible is seen in quite a different light.

## 2 Review of figures in IPCC reports: a predominance of projections of most likely impacts

Despite the argument set out above, it appears that the majority of graphs of future climate change impacts take the form of impact over time. By a rough count, the Working Group II contribution to the IPCC's Fifth Assessment Report (IPCC, 2014b; IPCC, 2014c) contains some 26 figures featuring graphs of impact over time (including eight where a time-dependent variable such as temperature increase or emissions may be considered a proxy for time on the x axis). It contains a similar number of figures featuring maps, which when presented individually show impacts at a fixed point in time, and when presented in time-series are equivalent to graphs of impact over time. The report has no figures containing graphs of probability over impact. It has only four figures showing graphs of impacts as probability over time (with proportion or frequency taken as proxies for probability), and two map sequences that can be interpreted in a similar way. Only two of these probability over time graphs, and one map sequence, clearly relate to relatively non-arbitrary thresholds of impact – defined in terms of their physical effect rather than in relation to their historical likelihood. The map sequence shows how the proportion of days in the year with temperatures above 40°C – when severe heat wave consequences are experienced – could increase over time in Australia. The two figures with graphs both relate to the risks to corals. One of these is reproduced here as Fig.4; it shows how the proportion of coral grids with degree heating months above threshold values for mass bleaching and mortality could change over time.

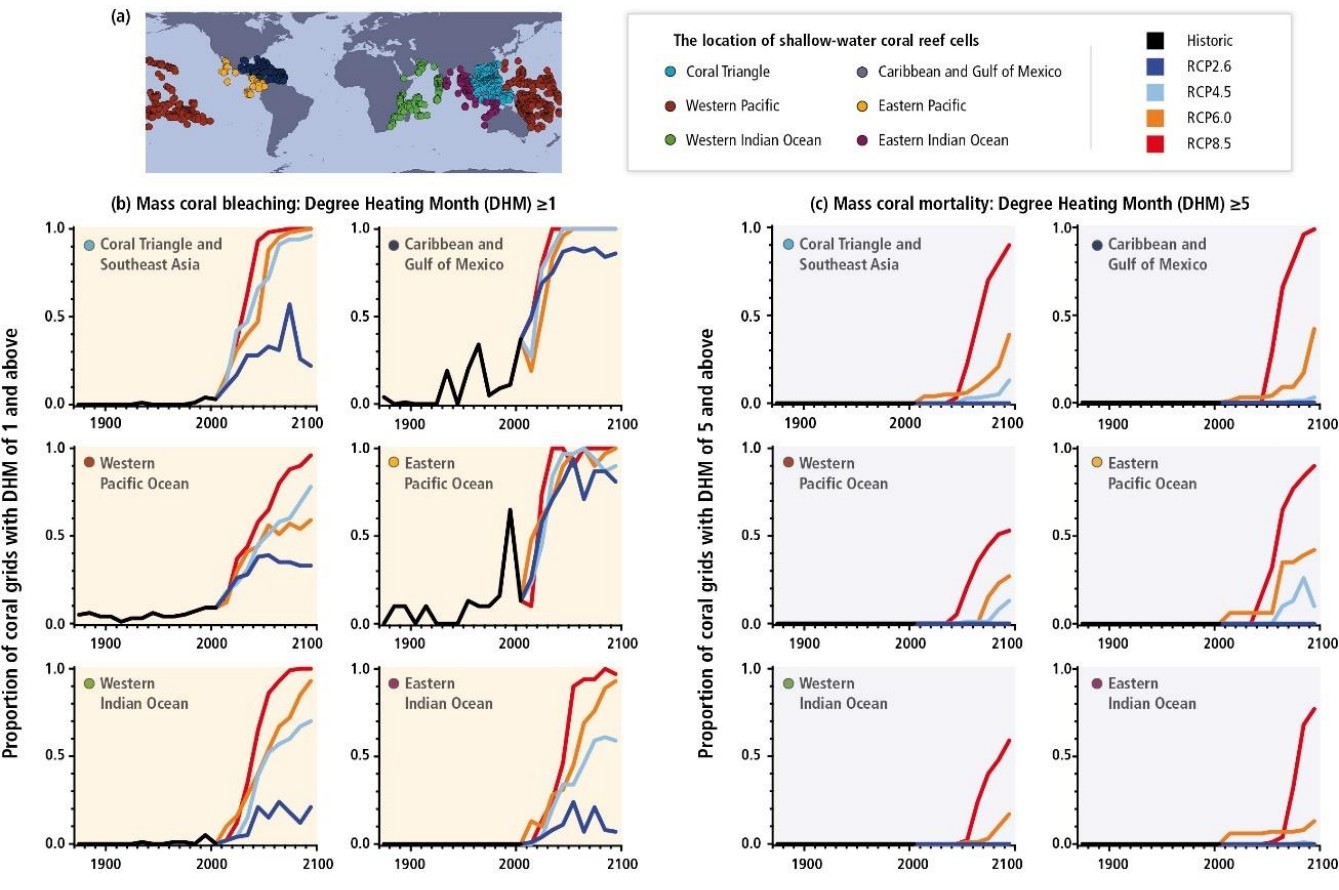

**Figure 4: The risk of mass coral bleaching / mortality presented in the form of probability / time[2] (Fig. 30-10 in IPCC, 2014d).**

---

[2] Original caption: Annual maximum proportions of reef pixels with Degree Heating Months (DHM, Donner et al., 2007) for each of the six coral regions (a, Figure 30-4b)—(b) DHM ≥1 (used for projecting the incidence of coral bleaching; Strong et al., 1997, 2011) and (c) DHM ≥5 (associated with bleaching followed by significant mortality; Eakin et al., 2010)—for the period 1870–2009 using the Hadley Centre Interpolated sea surface temperature 1.1 (HadISST1.1) dataset. The black line on each graph is the maximum annual area value for each decade over the period 1870–2009. This value is continued through 2010–2099 using Coupled Model Intercomparison Project Phase 5 (CMIP5) data and splits into the four Representative Concentration Pathways (RCP2.6, 4.5, 6.0, and 8.5). DHM were produced for each of the four RCPs using the ensembles of CMIP models. From these global maps of DHM, the annual percentage of grid cells with DHM ≥1 and DHM ≥5 were calculated for each coral region. These data were then grouped into decades from which the maximum annual proportions were derived. The plotted lines for 2010–2099 are the average of these maximum proportion values for each RCP. Monthly sea surface temperature anomalies were derived using a 1985–2000 maximum monthly mean climatology derived in the calculations for Figure 30-4. This was done separately for HadISST1.1, the CMIP5 models, and each of the four RCPs, at each grid cell for every region. DHMs were then derived by adding up the monthly anomalies using a 4-month rolling sum. Figure SM30-3 presents past and future sea temperatures for the six major coral reef provinces under historic, un-forced, RCP4.5 and RCP8.5 scenarios.

Figure 4, the exception that proves the rule, does a good job of communicating the risk: it makes it quite clear that on a high emissions pathway, it will be only a matter of time until most of the world's corals are extinguished.

A review of the IPCC's Special Report on Global Warming of 1.5°C (IPCC, 2018) suggests that this pattern has changed little
over the intervening four years. Impact over time graphs and maps still predominate, although temperature (a time-dependent variable) is used for the x axis rather than time itself – perhaps reflecting the report's purpose of demonstrating the differences between warming of 1.5°C and 2°C. While the proportion of probability over time figures has increased – to four out of seventeen relevant figures – only one of these clearly relates to a non-arbitrary physical threshold of impact. (This shows the fraction of global natural vegetation at risk of severe ecosystem change as a function of global mean temperature change – Fig.
3.16 in IPCC, 2018).

## 3 Discussion: the opportunity and need for more assessments of probability over time

There are many ways that the stock of probability over time assessments could be expanded. Non-arbitrary fixed points of impact can be defined in relation to several different kinds of thresholds:

- Physical: the height of sea level that puts an island under water;
- Biophysical: the degree of heat and humidity that exceeds human physiological tolerance*; or the temperature that exceeds a crop's tolerance*;
- Biochemical: the degree of acidity that prevents a shellfish from forming a shell;
- Socioeconomic: the quantity of per capita water resources required to meet basic human needs*; the daylight hours below dangerous levels of heat stress required for a subsistence agriculture lifestyle to remain viable; or the height
of sea level at which it becomes less costly to relocate a coastal city than to continue to protect it against flooding*;
- Experiential: the impact of a past event whose damage is well understood, e.g. a storm surge equal to that of Superstorm Sandy, or a European heat wave equal to that of 2003 (as illustrated in Christidis, Jones and Stott, 2015);
- Political: an agreed value, such as the 2°C target warming limit*.


Clearly, these different kinds of threshold vary in their objectivity, and in other ways likely to have implications for risk assessment. Socioeconomic thresholds may be more possible to overcome through adaptation than those defined by physical or biochemical properties alone. Experiential and political thresholds may have only transient value – useful for as long as the past event is remembered, or for as long as the policy holds. But what all these thresholds have in common is that they are
relevant to policymakers, because they are defined in terms of what it is that we collectively wish to avoid. Even a subjectively-defined threshold of impact, chosen for its social relevance, is less arbitrary in this sense than a fixed point of probability or of time.

The report 'Climate Change: A Risk Assessment' (King et al., 2015) demonstrated the feasibility of the probability over time approach by presenting some illustrative studies and discussion of the probabilities of exceeding a range of thresholds, including those marked with an asterisk in the list above, as a function of time (or time-dependent variables) for selected

locations and scenarios. These examples showed clearly that on certain pathways, the things we wish to avoid may become highly likely. The response to the report was encouraging: at the Chatham House climate change conference of 2015, attended by members of the international climate policy community, a presentation of the report's findings was voted the second-most valuable presentation of the conference, from a field of some thirty expert speakers.

The probability over time approach is not without its challenges. Most thresholds do not have a single 'correct' definition. For example, a temperature tolerance threshold may be biophysical, but how long above that temperature should be considered 'too long' is a matter of expert judgment. Many thresholds are specific to their location, meaning that a national or global risk assessment performed this way may need to be made up of a diverse range of quite distinct studies, rather than using the kind of consistent data that allows for aggregation. And for some thresholds, the probability of crossing them is not be quantifiable.

In these cases, the best approach may be to use qualitative descriptors of likelihood – such as the 'very unlikely', 'possible', and 'likely' used in the IPCC Working Group I's assessment of the likelihood of specific abrupt and irreversible changes in the climate system occurring during the twenty-first century (IPCC, 2013), or the 'low', 'moderate', 'substantial', 'severe', and 'critical' used by the UK government in its assessments of the likelihood of a terrorist attack (UK Government, 2019) – incorporating these into an assessment of how the risk will change as a function of time.


The experience of producing the abovementioned report suggested that the most significant obstacle to adopting the probability over time approach was not any difficulty with the science, but the need to start – *before doing any science* – with a subjective question: 'what is it that we wish to avoid?' Overcoming this obstacle is unlikely always to be as simple as asking policymakers what it is that they are most worried about. Without enough information to begin with, how can they know? An iterative

process of 'co-production' of the risk assessment may be ideal, but the responsibility for coordinating such a process is not clearly owned by any one party (De Meyer et al., 2018).

It is therefore worth considering how each part of the climate science community can contribute to bringing these assessments into being. Contributors to the IPCC's Working Group II could conduct risk assessments using impact thresholds of the kinds

described above. Some of these are likely to highlight thresholds of temperature or sea level rise at which non-linear increases in risk take place, and these thresholds could in turn be assessed in the form of probability over time by contributors to IPCC Working Group I. Working Group I might also be able to assess the risks of 'large-scale singular events' in this way. National climate change risk assessments could identify thresholds of impact of particular relevance to the country concerned. Researchers conducting extreme event attribution studies could run their models forward in time – to show how the probability

of crossing each newly-formed experiential threshold will continue to increase in the future. Research funders could play an influential role by structuring research calls in ways that require co-production with decision-makers, inter-disciplinary collaboration, and the application of general principles of risk assessment.

## 4 Conclusion

The risks of climate change can be understood more clearly when research starts by identifying what it is that we most wish to avoid, and then assesses its likelihood as a function of time. By providing a clearer picture of the overall scale of the risks of climate change, such assessments could help inform the most important decision of all: how much effort to put into reducing emissions.

The proposal is certainly not that all climate science should be done in this way. Fundamental research is indispensable; and there are many ways of communicating risks. The suggestion is that more research could be done expressly for the purpose of risk assessment than is done at present, and that a deliberate approach should be taken to identifying and assessing the biggest risks. Decarbonising the world economy will not be as easy as jumping out of a pot. That makes it all the more important that no opportunity is missed to communicate the severity of the risks to those in charge. The water is already

getting warm.

### Data availability

The data underlying Section 2 of this paper is stored with open access at the National Geoscience Data Centre (item 125176), and is available for download at https://www.bgs.ac.uk/services/ngdc/accessions/index.html#item125176

### Competing interests

The author declares that he has no conflict of interest.

### Acknowledgements

Especial thanks to Jason Lowe and Alistair Woodward, for sharing their knowledge and advice so generously, without which this work would not have been possible. Sincere thanks also to Sir David King, Chris Rapley, Kris de Meyer and Rowan Sutton, for their advice and support. Thanks also to the reviewers for their constructive and helpful comments.

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
