# Peer review of "Telling the boiling frog what he needs to know: why climate change risks should be plotted as probability over time"

_Geoscience Communication, 2019_

## Referee Comment (RC1) · Tebaldi (Referee) · 1 Feb 2019

I agree with all that is proposed in this piece, and I think it is important to try and persuade the assessment community to adopt multiple perspectives and multiple communication devices to increase the cogency and relevance of climate change communication. This proposal is one, and a valuable one, and I'd be happy to see this piece published.

Mine are not really requests for specific corrections or changes, I just want to point out a few things. In AR6 WG1 there is going to be a new chapter, chapter 12, which is attempting to facilitate exactly the type of cross-WG efforts the author is highlighting

and wishing for in this piece. The authors of Chapter 12 (I'm one) are trying to identify metrics and thresholds that have impact relevance, and whose changes can be assessed from observations and climate model output from the point of view of WG1 science (i.e. remaining strictly within the boundaries of hazard characterization). It is far from easy, and that may be perhaps better acknowledged in this piece. I think the type of approach that is called for here makes a lot of sense for specific impact analyses, but it is far from straightforward in big, global assessments (or even national ones) given the myriad of metrics/thresholds that are relevant to some sectors/regions and less relevant to others. But an attempt to at least identify some examples, in a multidisciplinary approach, is being made. So hopefully next time around some parts of the IPCC report will have more immediate resonance with what is being wished for here. One thing I would like to see discussed in a more nuanced way is the idea of global warming levels and targets. I appreciate the recognition that these are socio-political constructs: it is in my opinion an important distinction that needs to be made. I agree they have their "aspirational value" and as such they may be taken as the lens through which to carry out this type of communication, but I would like to see an explicit mention of the fact that there is no such thing as an absolute threshold below which we are safe and above which we are toast, at this global scale, and that makes the treatment of such thresholds very different from those that have a physical meaning. Gavin Schmidt had a good piece, a long time ago on RealClimate about this. http://www.realclimate.org/index.php/archives/2006/07/runaway-tipping-points-of-no-return/ I would be happy if this disclaimer was made explicit here, since at some level I feel as if focusing on this communication for such arbitrary threshold may sometime be less than productive. Lastly, just one point about the (idealized) examples of graphics presented in the paper. It may be good to point out that the type of deterministic relation shown by these single lines is very rarely (if ever) what we have, but that some fuzziness, or shading, communicating uncertainty will also be part of this type of display. I know it is obvious to many of us but all the examples happen to be devoid of uncertainty, and that is possibly misleading for a reader.

---

## Referee Comment (RC2) · Claudia Tebaldi (Referee) · 14 Feb 2019

Thank you for the careful and thoughtful reply, Simon, look forward to seeing this past the discussion stage!

---

## Referee Comment (RC3) · David Stainforth (Referee) · 20 Feb 2019

This discussion piece raises some important issues. I'd be happy to see it published.

I don't have any specific corrections or requested changes but I'll make a few comments.

The article argues that research should start by identifying what it is that we most want to avoid and then go on to assess the likelihood of this as a function of time. The author suggests that research funders have a key role to play in this endeavour "by structuring research calls in a way that mandates the inter-disciplinary approach and

the pre-research engagement with decision-makers". I see significant merit in adopting measures of this type and I support the author in these goals. Nevertheless there are risks in such an approach and I think the article would benefit from more nuance on certain points.

It is certainly true that left to its own devices climate change science can focus on issues that are more removed from societal relevance than they should be. However, the article seems to argue that climate change science has a simplistic role of answering the questions asked by society, as if it is simply a matter of turning a prediction handle. Pre-research engagement with decision makers is something I agree would have significant value but by structuring calls towards answering specific decision-relevant questions one can inadvertently encourage research which makes whatever assumptions are necessary to output an answer. What can get lost in this process is reflexion on whether the question asked is currently answerable with any reasonable degree of confidence. Are the assumptions justified? Researchers or research disciplines that perceive the need for more fundamental research in order to answer the question simply don't apply because they can't address the terms of the call; as a result the outputs can become biased and over confident. The approach treats research as simply an extended form of consultancy. It risks undermining consideration of which questions can currently be answered.

A related issue is the "fuzziness" referred to in Dr. Tebaldi's review. I strongly agree that such uncertainties are a critical part of communication but it is important that these uncertainties receive the consideration they require. By demanding that climate science answers specific questions in terms of time dependent probabilities the author is encouraging a situation where computer model ensembles are interpreted as providing such information. Maybe he has other ideas of how it would be provided but this is the most common and simplest approach. I have concerns that the approach recommended provides no pressure to robustly consider the reliability of such uncertainty assessments because this is difficult, time consuming and unlikely to be prioritised in

a funder's calls if they are principally driven by the need to answer specific questions.

Although I support most aspects of the recommendations in the paper they are not a panacea and if not addressed carefully could do more harm than good. Substantial rebalancing towards the approaches proposed would be extremely valuable but this article would be even more useful if the difficulties and risks were more thoroughly explored.

---

## Author Comment (AC2) · 22 Feb 2019

I'm grateful for these helpful comments.

I realise I should have made clear, and will do in amending the paper, that I'm not proposing that all climate science should be done this way. Clearly, if John Tyndall hadn't been interested in the molecular physics of radiant heat, and if a century's-worth of scientists after him hadn't pursued fundamental research into the physics of the climate and all the workings of the Earth's systems, then policymakers wouldn't have any questions about climate change for any scientists to answer. I have a deep respect for fundamental research, and recognise it as the foundation that makes any

policy-related inquiries possible. It's right that most science should be science-led.

I am in complete agreement with the reviewer's conclusion that a 'substantial rebalancing' towards the approach I'm advocating – not a complete conversion – is what would be appropriate and valuable. My concern is that within the vast body of research being done on climate science, not enough is being done with the express intent of informing risk assessment. To give an example: when I served on the UK's delegation to approve the Summary for Policymakers (SPM) of the AR5 Working Group 2 report on Impacts, I was surprised that mention of research finding that heat and humidity conditions could potentially exceed the limits of human physiological tolerance for heat stress was not being included in the SPM. I asked the lead authors whether they doubted the validity of this research finding. They said no, and that if anything, it probably understated the risk – but the rule was that a finding could only be mentioned in the SPM if it was supported by at least two separate pieces of research. And of the more than 12,000 scientific papers cited in the WG2 report, only one had asked whether a warming planet might at some time, in some places, become too hot for people. (Meanwhile, I found WG2 cited nine studies that looked at the impacts of climate change on ski resorts in Europe, and thirteen studies that considered the impacts on European grape growing and wine production.)

I agree that even when research is done for the purpose of informing risk assessment, it is unlikely to be a simple process of scientists answering policymakers' questions. It is more likely to need a dialogue to arrive at a shared understanding of what is possible, knowable and relevant to people's interests.

The point about uncertainty is really important. It's fair to say that the simplest approach to estimating probabilities of specific climate impacts will be through the use of computer model ensembles, and there are undoubtedly risks to this. The graphs of global temperature increase shown in Figures 2 and 3 of this paper were produced with models that did not incorporate earth system feedbacks; other work by the same authors [1] has shown that when these feedbacks are incorporated, the estimated probability

Interactive
comment

of a given degree of temperature increase is substantially higher, within a wide range of uncertainty. As the actuaries wrote in our climate change risk assessment report [2], models are imperfect, and a factor that is important in determining risk should never be excluded from consideration simply because it cannot be quantified.

Having said that, I don't think the proposed approach – of assessing the probability of crossing a non-arbitrary threshold of impact as a function of time – necessarily implies using a model, or even making quantified estimates. The IPCC AR5 WG1 Chapter 12 on 'long-term climate change: projections, commitments and irreversibility' included a table [3] showing the estimated likelihood of specific abrupt and irreversible changes in the climate system occurring during the 21st century. These likelihoods were estimated using expert judgment, and were expressed in terms such as 'very unlikely', 'possible', and 'likely' instead of being quantified. The table would have been consistent with the approach I'm proposing if it had indicated how these likelihoods could change as a function of time or global temperature increase. To the extent that such a time-dependent assessment could have been supported by the science, it would have been useful.

Unquantified and judgment-based estimates of probability are common in other areas of risk assessment. A good example is counter-terrorism, where the UK government uses five 'threat levels' to indicate the likelihood of a terrorist attack in the UK [4]. These are: Low - an attack is unlikely; Moderate - an attack is possible but not likely; Substantial - an attack is a strong possibility; Severe - an attack is highly likely; Critical - an attack is expected imminently.

In situations where there may be great danger, expert assessments such as these can be useful despite their deep uncertainties. I think the important thing is to use the best available information (whether quantified or not) to give the fullest possible assessment of the risk.

What I think this means for the recommendation about research funding is that it should
be driven not so much by the need to answer specific questions, but by the need to rigorously assess specific risks. If the purpose of a research call is understood to be risk assessment, then it should be carried out in accordance with general principles and best practice of risk assessment, which should include as a priority the robust consideration of uncertainty assessments.

As mentioned above, this is only proposed as an approach for some research calls, not for all of them. And I agree, it is not a panacea.

References

[1] Lowe, J.A. and Bernie, D., 'The impact of Earth system feedbacks on carbon budgets and climate response', Philosophical Transactions of the Royal Society, Vol 376 Issue 2119 (2018) https://royalsocietypublishing.org/doi/full/10.1098/rsta.2017.0263

[2] Hare, D., 'An Actuarial Perspective' in King, Schrag, Zhou, Qi and Ghosh, 'Climate Change: A Risk Assessment' (2015) pp20-21 http://www.csap.cam.ac.uk/links/13/1032/

[3] Table 12.4 on p1115 of Collins, M., R. Knutti, J. Arblaster, J.-L. Dufresne, T. Fichefet, P. Friedlingstein, X. Gao, W.J. Gutowski, T. Johns, G. Krinner, M. Shongwe, C. Tebaldi, A.J. Weaver and M. Wehner, 2013: Long-term Climate Change: Projections, Commitments and Irreversibility. In: Climate Change 2013: The Physical Science Basis. Contribution of Working Group I to the Fifth Assessment Report of the Intergovernmental Panel on Climate Change [Stocker, T.F., D. Qin, G.-K. Plattner, M. Tignor, S.K. Allen, J. Boschung, A. Nauels, Y. Xia, V. Bex and P.M. Midgley (eds.)]. Cambridge University Press, Cambridge, United Kingdom and New York, NY, USA. https://www.ipcc.ch/site/assets/uploads/2018/02/WG1AR5_Chapter12_FINAL.pdf

[4] UK Government, 'Terrorism and national emergencies' https://www.gov.uk/terrorism-national-emergency

---

## Short Comment (SC1) · 10 Mar 2019

This article points to an important improvement that can be made to the representation of risks related to climate change to ensure that risk assessment reports speak more clearly and directly to the concerns of policy makers. Below I will describe some additional psychological factors in support of the boiling frog's case, as well as give pointers for its successful implementation.

**An empirical imperative for testing alternative climate risk representations**

Studies in the psychology of risk have shown that its perception can be dramatically

affected by how a particular risk is presented or formulated (Slovic, 2010). Most formal risk assessments are highly cognitive endeavours. They rely on complicated mathematical analysis and are presented using formalisms that require expert knowledge to be understood. Beyond this aspect of risk known as risk-as-analysis, there are two related psychological dimensions to risk perception. Firstly, the risk-as-feeling hypothesis states that some risk problems generate an affective response while others might not. This affective response is sometimes an overt feeling of fear or anxiety, but often a more subdued "background" feeling (Loewenstein et al. 2001). Secondly, it appears that some cognitive representations of risk problems are intuitively evaluable while others are not (Cosmides Tooby, 2008). This means that they can be evaluated using fast, automatic cognitive processes, rather than requiring slower, deliberative reasoning. Both risk-as-feeling and intuitive evaluability are crucial in guiding people's day-to-day decision making. By extension, these psychological factors also affect how particular problems "jump out" at politicians and policy makers, and thus influence how they trade-off giving attention to some risks over others.

Whereas some risk representations may be naturally "intuitively evaluable" by many people, other representations only become so through prolonged exposure and expertise in a particular domain. The "Hazards - Vulnerability - Exposure" risk framework of the IPCC WG2 Fifth Assessment Report is a likely case in point (IPCC, 2014). Journalism research following the release of the report has shown that its risk framing was not picked up by the media - with the notable exception of some business media (Painter, 2015). Although the primary audience of the report is policy makers, the fact that the risk language used in the report's press release did not make it into articles about it suggests that it did not speak to the intuitions of the journalists involved. This can be seen as a proxy for other audiences outside of scientists and policy makers closely involved in the production of the WG2 report. These 'other audiences' may include policy makers in infrastructure, transport or treasury roles who will in future be affected by climate change risks, but whose professional background may lie in other domains such as economics or engineering. To make climate change risk representations gain

widespread traction outside of the narrow band of scientists and climate policy makers, they will need to be made more intuitively evaluable to those other audiences.

**How to make climate risk representations intuitively evaluable to policy makers?**

The degree to which a complex problem is intuitively evaluable is related to a person's "lived expertise" in that problem domain. Policy makers often have expertise in one or more specific domains, in which they may have a limited number of concrete decision concerns. These concerns can be called the 'risk currency' of the policy maker, i.e., the measures or quantities that fall within their remit to keep below certain levels or between certain boundaries. The author's story (in AC1) of "faeces floating the in the street" is one such case. In this particular example, the scientists involved found it straightforward to produce a graph in the risk currency of the policy makers, which was hence intuitively evaluable to them, given their expertise.

As both reviewers rightfully point out, defining non-arbitrary thresholds may not be feasible or appropriate for all climate science areas or for all policy concerns, nor may the probability of surpassing a non-arbitrary threshold be calculable. To give an example of the latter: when a group of scientists studying Atlantic Ocean currents recently briefed Members of the European Parliament about their research, the most pressing question the MEPs wanted answered was what the research meant for the flow of refugees across the Mediterranean! Here, the risk currency of the politicians is one that the science community may never be able to address directly.

What this example and the related concerns expressed by one of the reviewers (see comment RC3) make clear is the need for appropriate "co-production" of research questions between policy makers, scientists and research funders (De Meyer et al., 2018). Even if scientist may not be able to quantify the risk of refugees directly, they might find - in conversation with policy makers - intermediate ways to explain the possible multiplier effect of climate change on their concerns. To do this successfully for all policy questions, it will simply not be enough for policy makers to tell funders what

questions they want answered; nor can funders define in isolation what research is required; nor can scientists define what thresholds are relevant to the risk currencies of different policy makers without having meaningful conversations with them. The responsibilities for fixing this potential mismatch of knowledge production and knowledge requirements are fragmented, and will need joined-up and sustained efforts to resolve (ibid.). A first step to a solution is the recognition of the problem, and the author's contribution to this could not have come at a better time.

**References**

Cosmides, L., Tooby, J. (2008). Can a general deontic logic capture the facts of human moral reasoning? How the mind interprets social exchange rules and detects cheaters. Moral psychology, 1, 53-119.

De Meyer, K., Howarth, C., Jackson, A., Osborn, D., Rose, L., Rapley, C., Sharpe, S., and Welch, K. (2018). Developing Better Climate Mitigation Policies: Challenging current climate change risk assessment approaches. UCL Policy Commission on Communicating Climate Science; Report 2018-01.

IPCC, 2014: Summary for policymakers. In: Climate Change 2014: Impacts,Adaptation, and Vulnerability. Part A: Global and Sectoral Aspects. Contribution of Working Group II to the Fifth Assessment Report of the Intergovernmental Panel on Climate Change [Field, C.B., V.R. Barros, D.J. Dokken, K.J. Mach, M.D. Mastrandrea, T.E. Bilir, M. Chatterjee, K.L. Ebi, Y.O. Estrada, R.C. Genova, B. Girma, E.S. Kissel, A.N. Levy, S. MacCracken, P.R. Mastrandrea, and L.L.White (eds.)]. Cambridge University Press, Cambridge, United Kingdom and New York, NY, USA.

Loewenstein, G. F., Weber, E. U., Hsee, C. K., Welch, N. (2001). Risk as feelings. Psychological bulletin, 127(2), 267.

Painter, J. (2015). Taking a bet on risk. Nature Climate Change, 5(4), 286.

---

## Author Response (AR1)

**'Telling the Boiling Frog What He Needs to Know'**

**Review and discussion summary: comments, responses and changes to manuscript**

**Comment 1a**

I agree with all that is proposed in this piece, and I think it is important to try and persuade the assessment community to adopt multiple perspectives and multiple communication devices to increase the cogency and relevance of climate change communication. This proposal is one, and a valuable one, and I'd be happy to see this piece published. Mine are not really requests for specific corrections or changes, I just want to point out a few things. In AR6 WG1 there is going to be a new chapter, chapter 12, which is attempting to facilitate exactly the type of cross-WG efforts the author is highlighting and wishing for in this piece. The authors of Chapter 12 (I'm one) are trying to identify metrics and thresholds that have impact relevance, and whose changes can be assessed from observations and climate model output from the point of view of WG1 science (i.e. remaining strictly within the boundaries of hazard characterization). It is far from easy, and that may be perhaps better acknowledged in this piece. I think the type of approach that is called for here makes a lot of sense for specific impact analyses, but it is far from straightforward in big, global assessments (or even national ones) given the myriad of metrics/thresholds that are relevant to some sectors/regions and less relevant to others. But an attempt to at least identify some examples, in a multidisciplinary approach, is being made. So hopefully next time around some parts of the IPCC report will have more immediate resonance with what is being wished for here.

**Response 1a**

It's wonderful to know that work is going on to identify relevant impact thresholds for discussion in AR6. I agree identifying meaningful thresholds is not always easy. I think in most cases it takes a fair amount of expert judgment – for example, even in relation to a biophysical threshold such as the limit of human tolerance for heat stress, a judgment has to be made about how many hours of heat and humidity conditions above that limit is to be defined as an instance of 'crossing the threshold', around which estimates of probability (as a function of time) can be made. I'm happy to give that difficulty more acknowledgement. I suppose one way of saying this is that in almost no case is there a single 'correct' definition of the threshold, and this makes it difficult to choose the one that's most appropriate. At the same time, I'd argue that almost any reasonably defensible choice of threshold is better than no choice at all.

I also agree that many thresholds will be specific to their location, and so not applicable at a national or global scale. A good example of this was once recounted to me by a very experienced climate scientist in the US. After I had outlined to him my argument for taking a probability over time approach, he recalled one time his research team had done that. They had asked a city government what level of extreme rainfall they were worried about. The city government said it was X inches of rainfall over a period of Y hours (I don't recall the numbers, but they gave them), because this was the level that caused the city's sewer system to overflow, bringing faeces floating onto the streets. (That was their non-arbitrary threshold). The research team went away and came back with a plot of the probability of faeces floating in the streets as a function of time. The scientist told me that in all his years of advising this city government, he could not think of any other time when his advice had made such a strong impression. It was a great example of a non-arbitrary threshold, but perhaps not a widely replicable one.

Despite the local specificity of many thresholds, I think a collection of threshold-based risk assessments could add real value to either a national or a global assessment. For example, knowing that several parts of the world could exceed human tolerance for heat stress, that many places could exceed the temperature tolerances for several important crops, and that other places could fall below socioeconomic thresholds of minimum renewable water resources, can greatly enrich our understanding of the scale of the risks facing the world as a whole. There is no need for this information to be aggregated into global-scale metrics, or even for the same thresholds to be used for different regions, for this information to be useful. There may be an issue here that is beyond the scope of this paper, but I wonder whether there is a difference between the value of consistency of measurement for the purposes of advancing scientific understanding, and the value of diversity of measurement for the purposes of informing a risk assessment.

The main difficulty I've mentioned in the initial draft of this paper is the need to start – before doing any science – with a subjective question: 'what is it that we wish to avoid?' The point I'm making here is about the *order* in which things need to be done. In the heat stress example mentioned in the paper (and presented in this report http://www.csap.cam.ac.uk/projects/climate-change-risk-assessment/ ) the order was: first, have a policymaker define an issue of concern (environmental conditions exceeding human tolerance for heat stress); second, have an expert in the system of concern (human health) from the WG2 community define a meaningful threshold in relation to that system (X hours above Y degrees WBGT); and last, have an expert in the physical climate from the WG1 community estimate the probability of crossing the threshold as a function of time, for some regions where it looks likely to be a problem. Done this way round, it's quite feasible, but if you start with the WG1 science, it's almost impossible. I'd be interested to know whether you agree.

**Change 1a**

New text, p7 lines 7-11: The probability over time approach is not without its challenges. Most thresholds do not have a single 'correct' definition. For example, a temperature tolerance threshold may be biophysical, but how long above that temperature should be considered 'too long' is a matter of expert judgment. Many thresholds are specific to their location, meaning that a national or global risk assessment performed this way may need to be made up of a diverse range of quite distinct studies, rather than using the kind of consistent data that allows for aggregation.

**Comment 1b**

One thing I would like to see discussed in a more nuanced way is the idea of global warming levels and targets. I appreciate the recognition that these are sociopolitical constructs: it is in my opinion an important distinction that needs to be made. I agree they have their "aspirational value" and as such they may be taken as the lens through which to carry out this type of communication, but I would like to see an explicit mention of the fact that there is no such thing as an absolute threshold below which we are safe and above which we are toast, at this global scale, and that makes the treatment of such thresholds very different from those that have a physical meaning. Gavin Schmidt had a good piece, a long time ago on RealClimate about this. http://www.realclimate.org/index.php/archives/2006/07/runaway-tipping-points-ofno-return/ I would be happy if this disclaimer was made explicit here, since at some level I feel as if focusing on this communication for such arbitrary threshold may sometime be less than productive.

**Response 1b**

Regarding global warming thresholds, I'm happy to make an explicit mention of the fact that, as you say, 'there is no such thing as an absolute threshold below which we are safe and above which we are toast'. This is clearly true. I think thresholds with a physical basis are likely to be the most dangerous to cross, and so are particularly important to investigate. Socioeconomic thresholds may perhaps be more amenable to adaptation. What I've called 'experiential' thresholds may have only transient value – they are meaningful only for as long as people have strong memories of the original event. I think it would be going too far to attempt to rank different kinds of thresholds by how 'non-arbitrary' they are, but I agree it is important to distinguish between them, and useful to recognise their different characteristics.

The Gavin Schmidt piece is very good, and makes the point well that there is not just one tipping point in the climate system. It also illustrates the difficulty of defining meaningful non-arbitrary thresholds in relation to changes in the climate system itself (strictly within WG1 scope), as opposed to thresholds in relation to impacts on people (WG2 scope, to which everything

I've said above refers). However, it seems that even the relatively feasible examples are not always communicated as clearly as they could be. Schmidt gives the disappearance of the Greenland Ice Sheet as one of the best examples of a 'point of no return'. The IPCC's Special Report on 1.5°C did helpfully mention that 'these instabilities could be triggered at around 1.5°C to 2°C of global warming', but gave most prominence to the impact-over-time finding that the most likely global sea level rise at 2100 would be around 10cm lower at 1.5°C than at 2°C. It might have been more helpful to compare the *probability* of triggering disappearance of the GIS at (a sustained) 1.5°C, compared to that probability at 2°C. If illustrated by data such as those underlying figure 13.14c in AR5 WG1, this could perhaps have presented a very powerful picture of the difference in risk between the two levels of warming.

**Change 1b**

New text, p3 lines 25-26: There is no temperature threshold within which we are safe, and beyond which we are all cooked.

New text, p6 lines 26-30: Clearly, these different kinds of threshold vary in their objectivity, and in other ways likely to have implications for risk assessment. Socioeconomic thresholds may be more possible to overcome through adaptation than those defined by physical or biochemical properties alone. Experiential and political thresholds may have only transient value – useful for as long as the past event is remembered, or for as long as the policy holds. But what all these thresholds have in common is that they are relevant to policymakers, because they are defined in terms of what it is that we collectively wish to avoid.

**Comment 1c**

Lastly, just one point about the (idealized) examples of graphics presented in the paper. It may be good to point out that the type of deterministic relation shown by these single lines is very rarely (if ever) what we have, but that some fuzziness, or shading, communicating uncertainty will also be part of this type of display. I know it is obvious to many of us but all the examples happen to be devoid of uncertainty, and that is possibly misleading for a reader.

**Response 1c**

Finally, regarding shaded uncertainty bands, I agree this is worth mentioning. I thought I had done so, but hadn't. I'll add this in. A spectacular example of shaded bands showing the range of risk is figure 12.5 from AR5 WG1, which shows the uncertainty around long-term global temperature increase. Within the set of impact over time graphs, I have found these two (AR5 WG1 12.5, and 13.14 as mentioned above) exceptionally useful for communicating the extent of the risks, but have never met another non-scientist who was aware of them – probably because they were not included in the SPM. I can only guess they were excluded from the SPM because they were considered too long-term to be policy relevant, but arguably, since they show the largest risks, they are the most policy relevant of all. I think including long-enough time horizons for the largest risks to be visible in all SPMs could be helpful – and would be consistent with the general principle that a risk assessment should always consider the biggest risks.

**Change 1c**

New text, p2 lines 18-20: (Shaded bands illustrating uncertainty in impact can bring a broader range of risks into view, but still provide no guarantee that the biggest risks will be visible.)

**Comment 2a**

This discussion piece raises some important issues. I'd be happy to see it published. I don't have any specific corrections or requested changes but I'll make a few comments.

The article argues that research should start by identifying what it is that we most want to avoid and then go on to assess the likelihood of this as a function of time. The author suggests that research funders have a key role to play in this endeavour "by structuring research calls in a way that mandates the inter-disciplinary approach and the pre-research engagement with decision-makers". I see significant merit in adopting measures of this type and I support the author in these goals. Nevertheless there are risks in such an approach and I think the article would benefit from more nuance on certain points.

It is certainly true that left to its own devices climate change science can focus on issues that are more removed from societal relevance than they should be. However, the article seems to argue that climate change science has a simplistic role of answering the questions asked by society, as if it is simply a matter of turning a prediction handle. Pre-research engagement with decision makers is something I agree would have significant value but by structuring calls towards answering specific decision-relevant questions one can inadvertently encourage research which makes whatever assumptions are necessary to output an answer. What can get lost in this process is reflexion on whether the question asked is currently answerable with any reasonable degree of confidence. Are the assumptions justified? Researchers or research disciplines that perceive the need for more fundamental research in order to answer the question simply don't apply because they can't address the terms of the call; as a result the outputs can become biased and over confident. The approach treats research as simply an extended form of consultancy. It risks undermining consideration of which questions can currently be answered.

Although I support most aspects of the recommendations in the paper they are not a panacea and if not addressed carefully could do more harm than good. Substantial rebalancing towards the approaches proposed would be extremely valuable but this article would be even more useful if the difficulties and risks were more thoroughly explored.

**Response 2a**

I realise I should have made clear, and will do in amending the paper, that I'm not proposing that all climate science should be done this way. Clearly, if John Tyndall hadn't been interested in the molecular physics of radiant heat, and if a century's-worth of scientists after him hadn't pursued fundamental research into the physics of the climate and all the workings of the Earth's systems, then policymakers wouldn't have any questions about climate change for any scientists to answer. I have a deep respect for fundamental research, and recognise it as the foundation that makes any policy-related inquiries possible. It's
right that most science should be science-led.

I am in complete agreement with the reviewer's conclusion that a 'substantial rebalancing' towards the approach I'm advocating – not a complete conversion – is what would be appropriate and valuable. My concern is that within the vast body of research being done on climate science, not enough is being done with the express intent of informing risk assessment. To give an example: when I served on the UK's delegation to approve the Summary for Policymakers (SPM) of the AR5 Working
Group 2 report on Impacts, I was surprised that mention of research finding that heat and humidity conditions could potentially exceed the limits of human physiological tolerance for heat stress was not being included in the SPM. I asked the lead authors whether they doubted the validity of this research finding. They said no, and that if anything, it probably understated the risk – but the rule was that a finding could only be mentioned in the SPM if it was supported by at least two separate pieces of research. And of the more than 12,000 scientific papers cited in the WG2 report, only one had asked whether a warming planet
might at some time, in some places, become too hot for people. (Meanwhile, I found WG2 cited nine studies that looked at the impacts of climate change on ski resorts in Europe, and thirteen studies that considered the impacts on European grape growing and wine production.)

I agree that even when research is done for the purpose of informing risk assessment, it is unlikely to be a simple process of scientists answering policymakers' questions. It is more likely to need a dialogue to arrive at a shared understanding of what
is possible, knowable and relevant to people's interests.

**Change 2a**

New text, p8 lines 8-10: The proposal is certainly not that all climate science should be done in this way. Fundamental, policy-neutral research is indispensable; and there are many ways of communicating risks. The suggestion is that more research could be done expressly for the purpose of risk assessment than is done at present, and that a deliberate approach should be taken to
identifying and assessing the biggest risks.

**Comment 2b**

A related issue is the "fuzziness" referred to in Dr. Tebaldi's review. I strongly agree that such uncertainties are a critical part of communication but it is important that these uncertainties receive the consideration they require. By demanding that climate science answers specific questions in terms of time dependent probabilities the author is encouraging a situation where computer model ensembles are interpreted as providing such information. Maybe he has other ideas of how it would be provided but this is the most common and simplest approach. I have concerns that the approach recommended provides no pressure to robustly consider the reliability of such uncertainty assessments because this is difficult, time consuming and unlikely to be prioritised in a funder's calls if they are principally driven by the need to answer specific questions.

**Response 2b**

The point about uncertainty is really important. It's fair to say that the simplest approach to estimating probabilities of specific climate impacts will be through the use of computer model ensembles, and there are undoubtedly risks to this. The graphs of global temperature increase shown in Figures 2 and 3 of this paper were produced with models that did not incorporate earth system feedbacks; other work by the same authors [1] has shown that when these feedbacks are incorporated, the estimated probability of a given degree of temperature increase is substantially higher, within a wide range of uncertainty. As the actuaries wrote in our climate change risk assessment report [2], models are imperfect, and a factor that is important in determining risk should never be excluded from consideration simply because it cannot be quantified.

Having said that, I don't think the proposed approach – of assessing the probability of crossing a non-arbitrary threshold of impact as a function of time – necessarily implies using a model, or even making quantified estimates. The IPCC AR5 WG1 Chapter 12 on 'long-term climate change: projections, commitments and irreversibility' included a table [3] showing the estimated likelihood of specific abrupt and irreversible changes in the climate system occurring during the 21$^{st}$ century. These likelihoods were estimated using expert judgment, and were expressed in terms such as 'very unlikely', 'possible', and 'likely' instead of being quantified. The table would have been consistent with the approach I'm proposing if it had indicated how these likelihoods could change as a function of time or global temperature increase. To the extent that such a time-dependent assessment could have been supported by the science, it would have been useful.

Unquantified and judgment-based estimates of probability are common in other areas of risk assessment. A good example is counter-terrorism, where the UK government uses five 'threat levels' to indicate the likelihood of a terrorist attack in the UK [4]. These are:

- Low - an attack is unlikely
- Moderate - an attack is possible but not likely
- Substantial - an attack is a strong possibility
- Severe - an attack is highly likely
- Critical - an attack is expected imminently

In situations where there may be great danger, expert assessments such as these can be useful despite their deep uncertainties. I think the important thing is to use the best available information (whether quantified or not) to give the fullest possible assessment of the risk.

What I think this means for the recommendation about research funding is that it should be driven not so much by the need to answer specific questions, but by the need to rigorously assess specific risks. If the purpose of a research call is understood to be risk assessment, then it should be carried out in accordance with general principles and best practice of risk assessment, which should include as a priority the robust consideration of uncertainty assessments.

As mentioned above, this is only proposed as an approach for some research calls, not for all of them. And I agree, it is not a panacea.

**Change 2b**

New text, p7 lines 11-16: And for some thresholds, the probability of crossing them is not be quantifiable. In these cases, the best approach may be to use qualitative descriptors of likelihood – such as the 'very unlikely', 'possible', and 'likely' used in the IPCC Working Group I's assessment of the likelihood of specific abrupt and irreversible changes in the climate system occurring during the twenty-first century (IPCC, 2013), or the 'low', 'moderate', 'substantial', 'severe', and 'critical' used by the UK government in its assessments of the likelihood of a terrorist attack (UK Government, 2019) – incorporating these into an assessment of how the risk will change as a function of time.

**Comment 3a**

This article points to an important improvement that can be made to the representation of risks related to climate change to ensure that risk assessment reports speak more clearly and directly to the concerns of policy makers. Below I will describe some additional psychological factors in support of the boiling frog's case, as well as give pointers for its successful implementation.

An empirical imperative for testing alternative climate risk representations

Studies in the psychology of risk have shown that its perception can be dramatically affected by how a particular risk is presented or formulated (Slovic, 2010). Most formal risk assessments are highly cognitive endeavours. They rely on complicated mathematical analysis and are presented using formalisms that require expert knowledge to be understood. Beyond this aspect of risk known as risk-as-analysis, there are two related psychological dimensions to risk perception. Firstly, the risk-as-feeling hypothesis states that some risk problems generate an affective response while others might not. This affective response is sometimes an overt feeling of fear or anxiety, but often a more subdued "background" feeling (Loewenstein et al. 2001). Secondly, it appears that some cognitive representations of risk problems are intuitively evaluable while others are not (Cosmides Tooby, 2008). This means that they can be evaluated using fast, automatic cognitive processes, rather than requiring slower, deliberative reasoning. Both risk-as-feeling and intuitive evaluability are crucial in guiding people's day-to-day decision making. By extension, these psychological factors also affect how particular problems "jump out" at politicians and policy makers, and thus influence how they trade-off giving attention to some risks over others.

Whereas some risk representations may be naturally "intuitively evaluable" by many people, other representations only become so through prolonged exposure and expertise in a particular domain. The "Hazards - Vulnerability - Exposure" risk framework of the IPCC WG2 Fifth Assessment Report is a likely case in point (IPCC, 2014). Journalism research following the release of the report has shown that its risk framing was not picked up by the media - with the notable exception of some business media (Painter, 2015). Although the primary audience of the report is policy makers, the fact that the risk language used in the report's press release did not make it into articles about it suggests that it did not speak to the intuitions of the journalists involved. This can be seen as a proxy for other audiences outside of scientists and policy makers closely involved in the production of the WG2 report. These 'other audiences' may include policy makers in infrastructure, transport or treasury roles who will in future be affected by climate change risks, but whose professional background may lie in other domains such as economics or engineering. To make climate change risk representations gain widespread traction outside of the narrow band of scientists and climate policy makers, they will need to be made more intuitively evaluable to those other audiences.

How to make climate risk representations intuitively evaluable to policy makers?

The degree to which a complex problem is intuitively evaluable is related to a person's "lived expertise" in that problem domain. Policy makers often have expertise in one or more specific domains, in which they may have a limited number of concrete decision concerns. These concerns can be called the 'risk currency' of the policy maker, i.e., the measures or quantities that fall within their remit to keep below certain levels or between certain boundaries. The author's story (in AC1) of "faeces floating the in the street" is one such case. In this particular example, the scientists involved found it straightforward to produce a graph in the risk currency of the policy makers, which was hence intuitively evaluable to them, given their expertise.

As both reviewers rightfully point out, defining non-arbitrary thresholds may not be feasible or appropriate for all climate science areas or for all policy concerns, nor may the probability of surpassing a non-arbitrary threshold be calculable. To give an example of the latter: when a group of scientists studying Atlantic Ocean currents recently briefed Members of the European Parliament about their research, the most pressing question the MEPs wanted answered was what the research meant for the flow of refugees across the Mediterranean! Here, the risk currency of the politicians is one that the science community may never be able to address directly.

**Response 3a**

I'm grateful for these comments which provide additional insight into the issues raised in the paper.

I think the distinction between the information content of a risk assessment ('risk-as-analysis') and the presentation of a risk assessment ('risk-as-feeling' and 'intuitive evaluability') is a helpful one to make.

The main argument of my paper is that within the relatively narrow scope of the information content of the risk assessment ('risk-as-analysis'), there is room for improvement. If the variables of probability, impact and time have not been explored enough to bring the largest risks to light, then the risk assessment has omitted information that is likely to be of great relevance to the decision-maker. I propose identifying a non-arbitrary threshold of impact, and then assessing its probability over time, mainly because this seems likely to ensure that the largest risks are considered – and are not left hanging somewhere beyond the end of the x-axis or in the invisible margins of a shaded band of uncertainty. I believe that on this basis alone, there are advantages to this approach.

At the same time, I think it is quite possible that the probability over time approach also has an advantage in intuitive evaluability, and if so, this would be another argument in its favour. The discussion comment points out that 'The degree to which a complex problem is intuitively evaluable is related to a person's "lived expertise" in that problem domain'. This could explain the value of the 'experiential' threshold, as used in Christidis, Jones and Stott, 2015: once an extreme weather event has been experienced, it becomes part of the 'lived expertise' of all those it affected. Extreme event attribution studies use this effect to give greater social salience to the reporting of climate change in the present. Risk assessments using experiential thresholds of impact could do the same for the communication of possible climate changes of the future.

**Change 3a**

[No change made in response to this comment.]

**Comment 3b**

What this example and the related concerns expressed by one of the reviewers (see comment RC3) make clear is the need for appropriate "co-production" of research questions between policy makers, scientists and research funders (De Meyer et al.,
2018). Even if scientist may not be able to quantify the risk of refugees directly, they might find - in conversation with policy makers - intermediate ways to explain the possible multiplier effect of climate change on their concerns. To do this successfully for all policy questions, it will simply not be enough for policy makers to tell funders what questions they want answered; nor can funders define in isolation what research is required; nor can scientists define what thresholds are relevant to the risk currencies of different policy makers without having meaningful conversations with them. The responsibilities for fixing this
potential mismatch of knowledge production and knowledge requirements are fragmented, and will need joined-up and sustained efforts to resolve (ibid.). A first step to a solution is the recognition of the problem, and the author's contribution to this could not have come at a better time.

**Response 3b**

Finally: I agree with the point about the need for co-production of climate change risk assessments. The identification of a
meaningful non-arbitrary threshold of impact is often likely to be possible only through dialogue between scientists and decision-makers. It is useful to recognise that this dialogue does not necessarily happen by itself. Consequently, dedicated and deliberate processes may be needed to produce information that is appropriate for the purpose of risk assessment.

**Change 3b**

New text, p7 lines 20-23: Overcoming this obstacle is unlikely always to be as simple as asking policymakers what it is that
they are most worried about. Without enough information to begin with, how can they know? An iterative process of 'co-production' of the risk assessment may be ideal, but the responsibility for coordinating such a process is not clearly owned by any one party (De Meyer et al., 2018).

[revised manuscript text omitted]